# Unlocking the Land Capability and Soil Suitability of Makuleke Farm for Sustainable Banana Production

**Seome Michael Swafo * and Phesheya Eugine Dlamini ***

Department of Plant Production, Soil Science & Agricultural Engineering, School of Agriculture &
Environmental Sciences, University of Limpopo, Private Bag X1106, Sovenga 0727, Limpopo, South Africa
* Correspondence: 201600507@keyaka.ul.ac.za (S.M.S.); phesheya.dlamini@ul.ac.za (P.E.D.)

**Abstract:** Sub-Saharan Africa (SSA) is experiencing an increase in food insecurity, which is fueled by both high population growth and low agricultural productivity. Smallholder farmers are seriously affected by low soil fertility, land degradation, and poor agronomic management practices that reduce crop productivity. Therefore, there is a huge need for reliable soil information to support agricultural decision-making in smallholder farms to ensure sustainable agricultural production. However, most studies focused on land capability and soil suitability do not consider the spatial variability of soils and their inherent properties. The main objectives of this study were (1) to survey, classify and characterise soils at Makuleke farm in order to derive and map the land capability classes and (2) to quantify the physical and chemical properties of the soils in order to derive and map the suitability classes. A field survey and classification of soils led by transect walks complemented by auger holes revealed existential spatial variation of soils across the 12 ha banana plantation. The dominating soil forms in the plantation were Hutton, Westleigh, Glenrosa and Valsrivier. Land capability analysis revealed that 17% of the 12 ha portion of the farm had very high arable potential, while 60% had medium arable potential, 6% of the farm had low arable potential and 17% was considered non-arable. Subsequent soil suitability analysis revealed that 12% of the farm is highly suitable, 34% is moderately suitable, 38% is marginally suitable and 16% is permanently not suitable for banana production. The variable capability of the land and suitability of soils for banana production led to notable yield gaps. The in-depth description and quantification of the productive capacity of the land is pivotal to the farmers at Makuleke farm as it unlocks their true potential and such information is crucial to effectively manage the soil and utilize the land for sustainable banana production.

**Keywords:** land evaluation; land capability; soil suitability; smallholder farmers; soil spatial variability; soil information

## 1. Introduction

Food insecurity in Sub-Saharan Africa (SSA) is worsening, and it is underpinned by low crop output and high population expansion [1]. In South Africa, smallholder agriculture has been recognised as the vehicle through which the goals of poverty and rural development can be attained [2]. However, this type of farming is faced with numerous challenges, chief among which is soil degradation caused by unsustainable farming practices [3]. Soil degradation is often caused by the mismatch between land use and land potential, specifically using marginal lands for agriculture [4]. Moreover, the ever-increasing African population, which is directly proportional to an increase in the demand for food, makes the situation grimmer. This has consequently resulted in the active search for alternative approaches to agricultural production that do not only ensure that there is enough food at the table but do so sustainably [5].

One of the major challenges is to sustainably use the land to feed and sustain the burgeoning population [6]. Accordingly, there is a need to maintain and increase agricultural productivity in order to meet the increasing population pressure on arable land that fosters soil degradation, which threatens food production, especially in smallholder farming systems [7]. In recent times, there has been growing interest in better managing soils to underpin food security [8]. As such, the need for reliable soil information to support agricultural decision making has never been greater [9]. Improved management of soil resources and identification of the agricultural potential of soils is needed to prevent land degradation and stimulate crop production [10].

A better understanding of the factors limiting crop yields may provide a solution to reducing the existing yield gaps in smallholder farms [11]. The nexus between the quality of the soil and potential productivity is paramount in agriculture in pursuit of maximizing production and sustainability [12]. The assessment of the status of the soil and the capability of the land requires the proper establishment of a reference state-specific to each soil unit [13]. In this context, both land capability and soil suitability can be useful tools to ensure delineation of management zones aimed to improve agricultural productivity [14].

Land capability assessment is based on the inherited permanent physical properties of the land (i.e., slope, soil depth, texture and permeability), while soil suitability is based on soil properties (e.g., soil texture, organic carbon, nutrient availability and pH) which have greater impact on the growth of a specific crop [15]. An assessment of the factors influencing the capability and suitability of the land, such as the soil quality and climate, yields essential information on the potential of the land for agricultural use [16]. A majority of land capability and soil suitability studies do not consider the spatial variability of the soils and their inherent properties, yet this information is crucial for resolving site or location specific land management issues [17]. A thorough analysis of the soil spatial variability results in precise derivation of land capability and soil suitability classes [14]. In light of this, gathering precise site-specific information on land and soil resources can aid in identifying the limitations and potentials of these limited resources. The objectives of this study were (1) to survey, classify and characterise soils at Makuleke farm in order to derive and map the land capability classes and (2) to quantify the physical and chemical properties of the soils in order to derive and map the soil suitability classes.

## 2. Materials and Methods

### 2.1. Site Description and History

The Makuleke farm (30°56′16.3″ E and 22°51′31.9″ S) is situated in the Collins Chabane Local Municipality, Vhembe District, Limpopo Province in South Africa (Figure 1). The study was conducted on a 12 ha banana plantation portion of the farm. The farm experiences a humid subtropical climate with long summers and short winters characterised by rain and cool weather respectively [18]. The minimum and maximum temperature of the site is 12 °C and 30 °C respectively. The average annual temperature of the site is 21.7 °C, while the mean annual rainfall is 731 mm [19]. In terms of rainfall classification, the site falls under wet category.

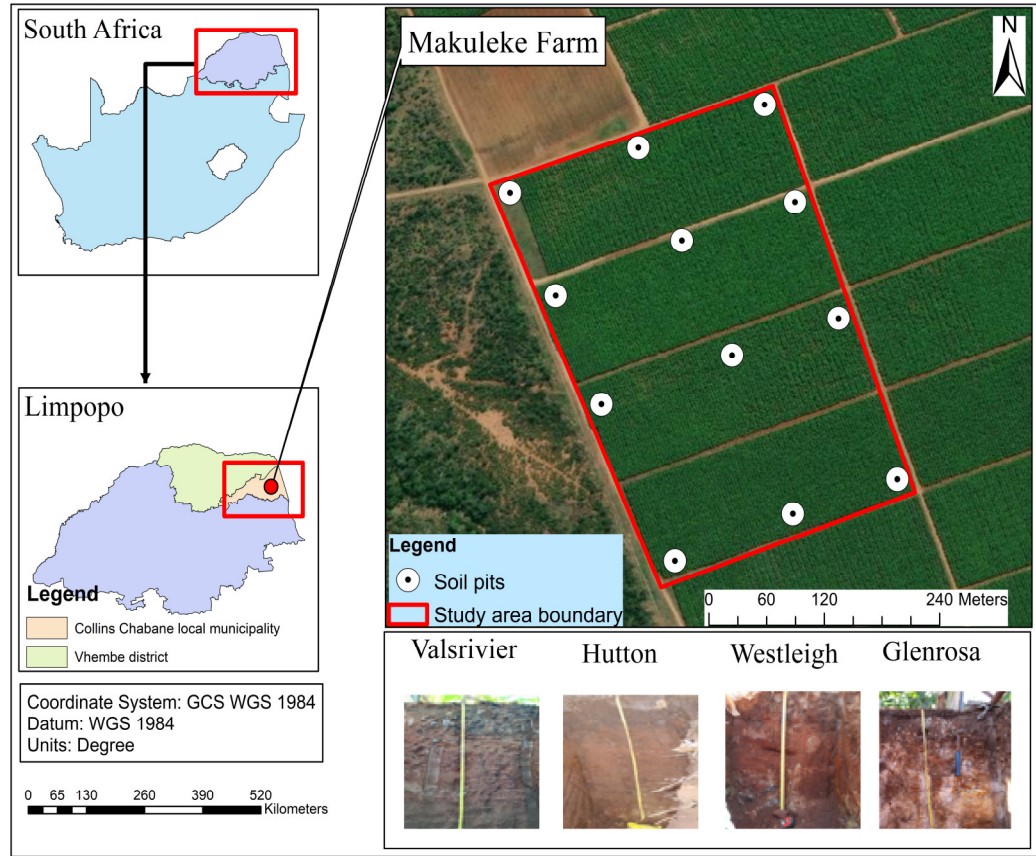

**Figure 1.** Location of Makuleke farm in the Collins Chabane Local Municipality, Vhembe District, Limpopo province, and photographs showing the dominant soil forms in the farm.

The Makuleke Farm (formerly known as Makuleke Irrigation Scheme) was established in 1985. The sole purpose of the scheme was to produce food (by planting crops) and to create jobs for Makuleke community members. This led to the establishment of Makuleke Farmers' Cooperative. The cooperative was first registered in 1991, comprising of 52 farmers. The farmers were producing maize, tomatoes and cabbage until the cooperative collapsed in 1998. It was only after the year 2000 that the farmers mobilised again to start cultivating crops. In 2007, Makuleke farm was producing maize, dry beans, and potatoes until December 2018, when they switched to banana fruit production. They switched to banana fruit because the crops (i.e., maize, dry beans and potatoes) were not doing well, characterised by low yield throughout the seasons. In terms of banana production, the farm can produce up to 56 tons per hectare of banana, which is below the average yield of bananas and that is 65 tons/ha.

### 2.2. Field Soil Survey and Classification

A field soil survey was conducted using transect walks complemented by auger holes to sub-divide the 12 ha portion of the farm into varied soil units [20]. This was done by grouping soils both to their properties and where they are located across the farm. At each defined soil unit, morphological features including soil colour, texture, and topographic attributes (i.e., slope gradient and elevation) were determined in the field. The focus on soil morphological properties, described in the field including soil texture, consistency, and structure is because they yield a significant benefit on the potential productivity of the soil [12].

Soil classification, which is linked to soil survey, was done to determine the morphological and pedological characteristics of the soil. To aid soil classification, pits were dug up to the limiting layer using an excavator. The dimensions of the dug soil

profiles were 1.5 m wide × 1.5 m long, and the depth was defined to the limiting layer. Twelve soil profile pits were sited, excavated, studied, described, and sampled. Soil profiles were described, and horizons delineated to determine the form, structure, and organization of soil material [21]. The thickness of each horizon and effective rooting depth of the soil profiles were determined using a measuring tape. Specifically, the effective rooting depth was determined based on the number of roots found within the depth of each opened soil pit. Soil colour for each diagnostic horizon was determined by matching a freshly broken soil fragment to the Munsell colour chart both in dry and moist state. Soil permeability and slope percentage were determined following the methodology by [22]. Each soil profile was georeferenced using a portable handheld global positioning system (GPS) (Model Garmin 12 L).

### 2.3. Collection of Soil Samples in the Field

The soil samples were collected from the 12 dug soil profiles. At each dug pit, soil samples were collected at the 30 cm depth interval on three faces. The three faces of the pit served as replicates. This means that from each pit, three soil samples were collected at the 30 cm depth interval. As such, a total of 36 samples were collected representing the 12 dug profiles across the 12 ha plantation. The soil samples were collected using a spade and a geological hammer. The collected samples were bagged and labelled according to the georeferenced pits and replicate number. The soil samples were carefully handled and then taken to the laboratory for preparation and analysis.

### 2.4. Preparation and Laboratory Analysis of Soil Physicochemical Properties

In the laboratory, soil samples were air-dried, crushed, and then passed through a 2 mm sieve in the laboratory prior to soil physical and chemical analyses. Particle size distribution of sand, silt, and clay content was determined using the hydrometer method [23]. Soil organic carbon was determined using the Walkley-Black Method [24]. Soil pH (KCl) was determined using 1:25 1 mol dm$^3$ KCl ratio suspensions on mass-based methods respectively and read with a glass electrode pH meter [25]. Soil phosphorus (P) was determined on a 2-mL aliquot filtrate using a modification of the Murphy and Riley [25]molybdenum blue procedure. Soil calcium (Ca), magnesium (Mg) and potassium (K) were determined by atomic absorption (using an air-acetylene flame) on a 5 mL aliquot of the filtrate after dilution with 20 mL de-ionized water [26]. Total nitrogen (TN) was analyzed by an automated Dumas dry combustion method using a LECO TruSpec CN (LECO Corporation, Michigan, USA; Matejovic, 1996) [25].

### 2.5. Derivation of Land Capability and Soil Suitability Classes

The land capability classes were derived using the concepts and principles of the FAO Framework for Land Evaluation [27], but adapted to South African conditions by [22]. The physical land attributes and morphological characteristics of the soil that were used to derive the capability classes include slope percentage, soil texture, soil permeability, and effective rooting depth [22]. Once all the physical characteristics of the land and morphological features of the soil were gathered, a land evaluation criterion was followed to assess the capability of the land for arable agriculture. Land capability classes of the studied farm were derived using the agricultural assessment framework developed by [22].

The FAO framework for land evaluation [27] coupled with the guidelines for rainfed agriculture [28] was used to determine the suitability of the soil at Makuleke farm. The criteria proposed by [29] and [15] for crop suitability with degrees of limitations were adopted and logically categorized based on soil site characteristics for highly suitable (S1), moderately suitable (S2), marginally suitable (S3), currently (N1), and permanently not suitable (N2) classes. Soils that fall under S1 have no substantial limitations to sustained application for a specific use. Moderately suitable (S2) soils have moderate severe

limitations for sustained application of a particular use, while soils in the S3 category have severe limitations for sustained application of a specific use. Currently not suitable (N1) soils have limitations which may be manageable in time, however the limitations are so severe that they impede successful sustained use of the land for a given use. Permanently not suitable (N2) soils have severe limitations that eliminate any chance of being successfully used in the intended way [27]. Soil suitability classification was done by matching plant growth requirements of banana with agro-climatic, soil properties (soil texture, pH, and SOC), and land physical characteristics (i.e., topography) (Table 1) [15,29,30].

**Table 1.** Soil site criteria determination for banana fruit.

| Soil Site Characteristics | Class, Degree of Limitation and Rating Scale | | | | |
|---|---|---|---|---|---|
| | **S1** | **S2** | **S3** | **N1** | **N2** |
| Climatic Regime (c) | | | | | |
| Mean temperature in growing season (°C) | 26–33 | 34–36; 24–25 | 37–38 | >38 | |
| Topography (t) | | | | | |
| Slope (%) | 0–2 | 2–4 | 4–8 | 8–16 | - | >16 |
| Wetness (w) | | | | | |
| Drainage | Good | Well drained | Moderately drained | Poorly drained | Very poorly drained |
| Physical soil characteristics (s) | | | | | |
| Texture/structure. | L, Cl, Scl, Sil | Sicl, Sc, C (<45%) | C (>45%), Lic, sl | Is, s | |
| Soil depth (M) | >1.25 | 1.25–0.75 | 0.5–0.75 | <0.5 | |
| Soil fertility characteristics (f) | | | | | |
| Base saturation (%) | >50 | 50–35 | 35–20 | <20 | - | - |
| Sum of basic cations (cmol (+)/kg soil) | >6.5 | 6.5–4 | 4–2.8 | - | - | - |
| pH | 6.0–5.4 | 5.4–5.0 | 5.0–4.8 | 4.8–4.1 | <4.1 | - |
| Organic carbon (%) | >2.4 | 2.4–1.5 | 1.5–0.8 | <0.8 | - | - |

S1, Highly suitable; S2, Moderately suitable; S3, Marginally suitable; L, Loam; cl, Clay; scl, Sandy clay loam; sil, Silt loam; Sc, sandy clay; C, clay; Ls, Loamy sand; S, Sand; N1, Currently not suitable; N2, Permanently not suitable.

### 2.6. Generation of Soil Form, Land Capability and Soil Suitability Maps

The soil form (also referred to soil type), land capability and soil suitability maps were generated using Google Earth pro (Google earth, 2022, Keyhole, Inc., Mountain View, CA, USA) and ArcGIS 10.8.1 software (ESRI, Redlands, CA, USA). Firstly, the coordinates of the 12 profile pits were used to demarcate the location of each profile pit using the "add placemark" tool in Google Earth. Each placemark was labelled according to the name of the soil form found at that particular profile pit. Once all the 12 placemarks were inserted and labelled, the "add path" tab was used to join placemarks of the same soil form. The soil characteristics determined from the profile pits were used to establish the mapping units. The "add polygon" tool was used to create a polygon using the joined placemark of Westleigh, Valsrivier, Hutton and Glenrosa soil forms. Then the polygon was digitised to create the shape of each soil form. The polygon was named according to soil form and then saved as a KML layer.

Secondly, in ArcMap, a conversion tool "From KML" was used to convert the KML layers of the polygons from Google Earth and thereafter saved as a shapefile. Once all the shapefiles of the four soil forms were generated, a spatial distribution map was produced by "checking" all the shapefiles on the same data frame. The "add data" tool was used to insert the base map of the Makuleke farm. Thereafter shapefiles of South African provinces, districts and local municipalities were used to extract the Limpopo province, Vhembe district and Collins Chabane local municipality using the "select tool". In case of land capability and soil suitability maps, each profile pit was renamed according to a

derived land capability and soil suitability class. Then they were mapped following the procedure used to map the soil forms.

## 3. Results

### 3.1. Pedological and Morphological Characteristics of the Soils Underlying the 12 ha Banana Plantation

Soil morphology describes and measures a wide range of characteristics of the soil within the numerous soil horizons [31]. It deals with the form, structure, kind and arrangement of the soil material within the horizons. In the study area (Makuleke farm), four soils were identified and classified as Valsrivier (4.5 ha), Westleigh (1.4 ha), Hutton (4.05 ha) and Glenrosa (2.05 ha) (Figure 2) (Soil Classification Working Group, 2018). The classified soils at the farm, namely Valsrivier, Westleigh, Hutton and Glenrosa are well known as Lixisols, Plinthosols, Cambisols and Leptosols, respectively from the World Reference base for soil resources [32].

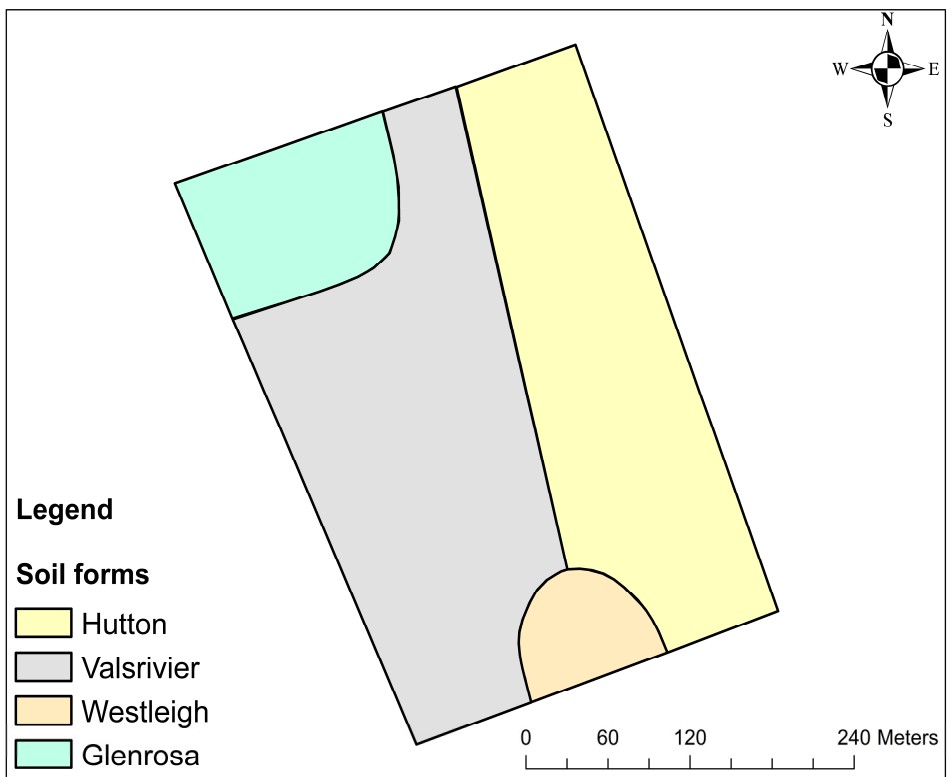

**Figure 2.** Spatial distribution of soil forms across the 12 ha banana plantation.

The Valsrivier soil (Lixisol) covered 38% (equivalent to 4.5 ha) of the 12 ha banana plantation. This soil form was mainly found at the footslope and middleslope positions of the farm (Table 2). At the footslope, the soil was characterised by a dark reddish colour (10R 2.5/1), 0.30 m thick orthic A horizon underlain by dusky red colour (2.5YR 3/2), 1.2 M thick pedocutanic B horizon. The permeability of the soil ranged from 1–3 s, with a clay content of 19% and a slope class of 0–3%. The effective rooting depth was 0.30 m and the total depth of the pit was 1.50 m. At the middleslope, Valsrivier soil form was characterised by a dark reddish brown (5YR 3/3) colour, 0.31 m thick orthic A horizon underlain by a dusky red (10R 3/3), 0.60 m thick pedocutanic B horizon. The permeability of this soil ranged from 1–3 s, with an average clay content of 29% and a slope class ranging from 3–8%. The effective rooting depth of the soil was 0.30 m and the total soil depth was 0.91 m. These soil forms fall under the duplex soil group [33]. They are enriched with clay in the subsoil, which results in strong blocky and cutanic character.

**Table 2.** Pedological and morphological characteristics of soils at Makuleke farm located in the footslope and middleslope position.

| Transect No. | Pit No. | Topsoil Name | Colour (Topsoil) | Subsoil Name | Colour (Subsoil) | TSD (m) | ERD (mm) | Soil Form | Permeability (s) | Slope (%) | Terrain Unit | Particle Size Distribution | | | Texture Class |
|---|---|---|---|---|---|---|---|---|---|---|---|---|---|---|---|
| | | | | | | | | | | | | Clay (%) | Silt (%) | Sand (%) | |
| 1 | 1 | Orthic A | 10R 2.5/1 Reddish black | Pedocutanic B | 2.5YR 3/2 Dusky Red | 1.5 | 200–300 | Valsrivier | 1–3 | 0–3 | Footslope | 19 | 26 | 55 | Sandy loam |
| | 2 | Orthic A | 5YR 3/4 Dark Reddish Brown | Soft Plinthic B | 2.5YR 4/6 Red | 1.02 | 0–200 | Westleigh | 1–3 | 0–3 | Footslope | 29 | 25 | 46 | Sandy clay loam |
| | 3 | Orthic A | 5YR 3/4 Dark Reddish Brown | Red Apedal B | 5YR 3/4 Dark Reddish Brown | 1.35 | 200–300 | Hutton | 1–3 | 0–3 | Footslope | 41 | 33 | 26 | Clay |
| 2 | 1 | Orthic A | 7.5YR 3/4 Dark Brown | Pedocutanic B | 7.5YR 3/3 Dark Brown | 1.32 | 300–500 | Valsrivier | 4–8 | 0–3 | Footslope | 25 | 27 | 48 | Sandy clay loam |
| | 2 | Orthic A | 7.5YR 3/4 Dark Brown | Pedocutanic B | 2.5YR 4/4 Reddish Brown | 3.01 | 200–300 | Valsrivier | 4–8 | 0–3 | Footslope | 41 | 33 | 26 | Clay |
| | 3 | Orthic A | 10R 3/3 Dusky Red | Red Apedal B | 5YR 4/6 Yellowish Red | 1.16 | 200–300 | Hutton | 4–8 | 0–3 | Footslope | 39 | 32 | 29 | Clay loam |
| 3 | 1 | Orthic A | 5YR 3/3 Dark Reddish Brown | Pedocutanic B | 10R 3/3 Dusky Red | 0.907 | 200–500 | Valsrivier | 1–3 | 4–8 | Middleslope | 29 | 27 | 44 | Clay loam |
| | 2 | Orthic A | 5YR 3/3 Dark Reddish Brown | Pedocutanic B | 2.5YR 3/3 Dark Reddish Brown | 1.35 | 0–200 | Valsrivier | 1–3 | 4–8 | Middleslope | 33 | 33 | 34 | Clay loam |
| | 3 | Orthic A | 5YR 3/4 Dark Reddish Brown | Red Apedal B | 2.5YR 3/4 Dark Reddish Brown | 1.12 | 200–300 | Hutton | 1–3 | 4–8 | Middleslope | 33 | 31 | 36 | Clay loam |
| 4 | 1 | Orthic A | 5YR 3/3 Dark Reddish Brown | Lithocutanic B | 2.5YR 4/4 Reddish Brown | 1.2 | 0–200 | Glenrosa | 1–3 | 4–8 | Middleslope | 21 | 17 | 62 | Sandy clay loam |
| | 2 | Orthic A | 5YR 3/3 Dark Reddish Brown | Pedocutanic B | 5YR 3/4 Dark Reddish Brown | 1.3 | 0–200 | Valsrivier | 1–3 | 4–8 | Middleslope | 25 | 33 | 42 | Loam |
| | 3 | Orthic A | 7.5YR 3/3 Dark Brown | Red Apedal B | 5YR 3/4 Dark Reddish Brown | 1.1 | 200–300 | Hutton | 1–3 | 4–8 | Middleslope | 39 | 32 | 29 | Clay loam |

TSD, total soil depth; ERD, effective rooting depth.

The Westleigh soil form (Plinthosol) covered 12% (1.4 ha) of the Makuleke farm and was mainly found in the footslope position (Table 2). This soil was characterised by a dark reddish (5YR 3/4), 0.34 m thick orthic A horizon underlain by a red (2.5YR 4/6), 0.68 m thick soft plinthic B horizon. The permeability of the soil ranged from 1–3 s, with a clay content of 29% and a slope class of 0–3%. The effective rooting depth was 0.20 m and total soil depth of 1.02 m. The Westleigh soil form falls under the plinthic soil group [33]. The plinthic horizon has 25% by volume or more of an iron-rich, humus-poor mixture of kaolinitic clay with quartz, which changes irreversibly to a hard mass or to irregular aggregates on exposure to repeated wetting and drying with free access to oxygen [33].

The Hutton soil (Cambisol) covered 34% (4.05 ha) of the farm. It was found at the footslope and middleslope positions (Table 2). At the footslope, this soil form was characterised by a dark reddish brown (5YR 3/4), 0.34 m thick orthic A horizon underlain by a dark reddish brown (5YR 3/4), 1 m thick red apedal B horizon. The permeability of the soil ranged from 1–3 s, with a slope class of 0–3%. The effective rooting depth of the soil was 0.30 m and the total soil depth was 1.35 m, with a clay content of 36%. At the middleslope position, the Hutton soil was characterised by a dark reddish brown (5YR 3/4), 0.35 m thick orthic A horizon underlain by a dark reddish brown (2.5YR 3/4), 0.77 m thick red apedal B horizon. The permeability of the soil ranged from 1–3 s, with a slope class of 3–8%. The effective rooting depth and total depth of the soil was 0.3 m and 1.12 m respectively. The Hutton soil falls under the Oxidic soil group [33]. An overriding feature of Oxidic soils is uniformity of the B horizon colour. Oxidic soils have a B horizon that is uniformly coloured with red and/or yellow oxides of iron [33].

The Glenrosa (Leptosols) soil form covered 16% (2.05 ha) of the farm, and was found at the middleslope position. This soil was characterised by a dark reddish brown (5YR 3/3) colour, 0.05 m thick Orthic A surface horizon underlain by a reddish brown (2.5YR 4/4), 1.15 m thick Lithocutanic B horizon. The permeability of the soil ranged from 1–3 s, with a slope class of 3–8%. The total depth of the horizon was 1.2 m and the effective rooting depth was 0.20 m. This soil type falls under the Lithic soil group. The prevailing characteristic of lithic soils is their resemblance with the underlying parent rock [33].

### 3.2. Chemical Properties of the Soils across the 12 ha Banana Plantation

The chemical properties of the classified soil pits across the banana plantation are depicted in Table 3. In transect one at the footslope position, Valsrivier soil form had soil P, K Ca and Mg content of 32 mg/kg, 147 mg/kg, 2662 mg/kg and 573 mg/kg respectively (Pit 1). The pH of this soil (Valsrivier) was 5.20 while OC and N was 1.6% and 0.04% respectively. Westleigh soil form had a P, K, Ca and Mg content of 19 mg/kg, 157 mg/kg, 1355 mg/kg and 340 mg/kg respectively (Pit 2). The pH of Westleigh was 5.22 with OC content of 1.4% and N of 0.08%. Soil P, K, Ca and Mg content of Hutton soil was found to be 12 mg/kg, 82 mg/kg, 1649 mg/kg and 429 mg/kg respectively (Pit 3). Moreover, the pH of Hutton was found to be 5.31 with the OC % of 0.5 and N % of 0.03. In transect two at the footslope, on average Valsrivier soil had a P, K, Ca and Mg of 28 mg/kg, 240 mg/kg, 1934 mg/kg, and 579 mg/kg respectively (Pit 1 and 2). The average pH, OC and N of the soil (Valsrivier) was 5.02, 1.4% and 0.05% respectively. The soil P, K, Ca and Mg content of Hutton soil was 31 mg/kg, 175 mg/kg, 1539 mg/kg and 375 mg/kg respectively (Pit 3). This Hutton soil had a pH, OC and N of 5.05, 1.6% and 0.09% respectively.

**Table 3.** Chemical properties of the soils from the soil pits across the 12 ha banana plantation.

| Transect No. | Pit No. | Soil Form | Slope % | Terrain Unit | P | K | Ca | Mg | pH | OC | N |
|---|---|---|---|---|---|---|---|---|---|---|---|
| | | | | | (mg/kg) | | | | (KCl) | % | |
| 1 | 1 | Valsrivier | 0–3 | FS | 32 | 147 | 2662 | 573 | 5.2 | 1.6 | 0.04 |
| | 2 | Westleigh | 0–3 | FS | 19 | 157 | 1355 | 340 | 5.22 | 1.4 | 0.08 |
| | 3 | Hutton | 0–3 | FS | 12 | 82 | 1649 | 429 | 5.31 | 0.5 | 0.03 |
| 2 | 1 | Valsrivier | 0–3 | FS | 24 | 174 | 1843 | 751 | 5.06 | 1.3 | 0.03 |

| | | | | | | | | | | | |
|---|---|---|---|---|---|---|---|---|---|---|---|
| | 2 | Valsrivier | 0–3 | FS | 31 | 305 | 2025 | 406 | 4.97 | 1.5 | 0.06 |
| | 3 | Hutton | 0–3 | FS | 31 | 175 | 1539 | 375 | 5.05 | 1.6 | 0.09 |
| 3 | 1 | Valsrivier | 4–8 | MS | 21 | 106 | 1790 | 498 | 5.42 | 0.9 | 0.03 |
| | 2 | Valsrivier | 4–8 | MS | 18 | 112 | 1725 | 425 | 5.04 | 1.7 | 0.04 |
| | 3 | Hutton | 4–8 | MS | 23 | 206 | 1805 | 501 | 5.27 | 1.2 | 0.03 |
| 4 | 1 | Glenrosa | 4–8 | MS | 8 | 175 | 1212 | 327 | 4.62 | 1.1 | 0.05 |
| | 2 | Valsrivier | 4–8 | MS | 37 | 254 | 2424 | 361 | 5.25 | 1.7 | 0.08 |
| | 3 | Hutton | 4–8 | MS | 29 | 147 | 1862 | 477 | 5.33 | 1.2 | 0.05 |

FS, Footslope; MS, middleslope; P, Phosphorus; K, Potassium; Ca, Calcium; Mg, Magnesium; OC, Organic carbon; N, nitrogen.

In transect three at the middleslope position, on average Valsrivier soil had a P, K, Ca and Mg content of 19 mg/kg, 109 mg/kg, 1758 mg/kg and 462 mg/kg respectively (Pit 1 and 2). The average pH, OC and N was 5.23, 0.5% and 0.04% respectively. The Hutton soil had a P content of 23 mg/kg while the K, Ca and Mg content was 206 mg/kg, 1805 mg/kg and 501 mg/kg respectively (Pit 3). The pH, OC and N of this soil was found to be 5.27, 1.2% and 0.03% respectively. In transect four at the midslope, Glenrosa soil form had a soil P, K, Ca and Mg content of 8 mg/kg, 175 mg/kg, 1212 mg/kg and 237 mg/kg respectively (Pit 1). This soil had a pH of 4.62, with the OC and N of 1.1% and 0.05% respectively. The soil P, K, Ca and Mg of Valsrivier soil form was found to be 37 mg/kg, 254 mg/kg, 2424 mg/kg and 361 mg/kg respectively (Pit 2). The pH, OC and N of this soil form (Valsrivier) was found to be 5.25, 1.7% and 0.08% respectively. The Hutton soil form had a P, K, Ca and Mg content of 29 mg/kg, 147 mg/kg, 1862 mg/kg and 477 mg/kg. The pH of Hutton was found to be 5.33 while the OC and N content was 1.7% and 0.08% respectively.

### 3.3. Land Capability Classification for Arable Farming

The farm showed variable capability use classes, ranging from class I to VI (Figure 3). The land capability class I, III, IV, and VI covered 17%, 61%, 6% and 16% respectively of the farm. Lands in class I, III, and IV are referred to as arable with class I having none or few limitations. Class III and IV lands have moderate and severe limitations respectively, that constrain their use [21]. Class VI lands are described as not suitable for cultivation of crops, as their limitations hinder the growth of crops. Class I falls under Hutton (occupied 17%). Class III falls under Westleigh, Hutton (occupied 17%) and Valsrivier (occupied 32%), while Class IV falls under Valsrivier (occupied 6%). Class VI lands fall under Glenrosa soil. Classes I to IV falls under arable and class IV under low arable potential while class VI falls under non arable potential [22].

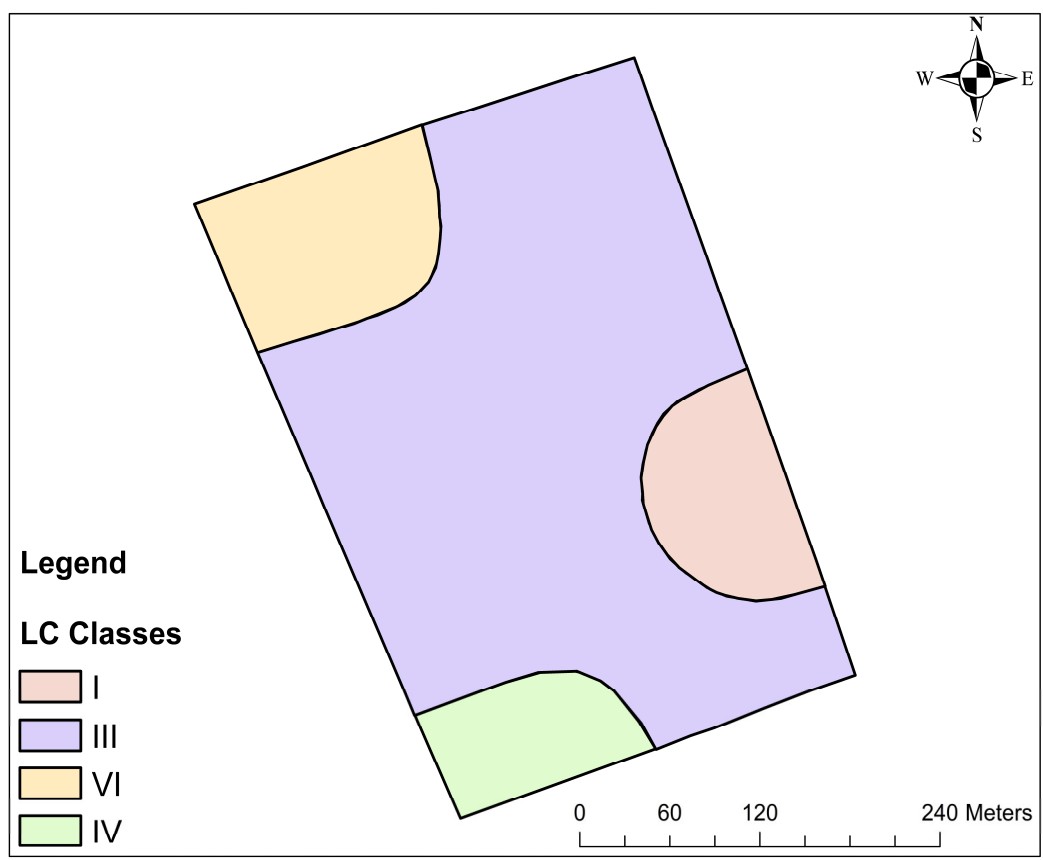

**Figure 3.** Spatial distribution of land capability classes across the 12 ha banana plantation.

*3.4. Soil Site Suitability for Banana Production*

Figure 4, shows the soil suitability classes of the farm utilised for banana production. Soil site suitability assessment for banana revealed that 12%, 34%, 37% and 16% is highly suitable (S1), moderately suitable (S2), marginally suitable (S3), and permanently not suitable (N2) respectively for banana cultivation. The moderately suitable portion of the farm has slight to moderate limitations caused by slope, texture, pH and OC for banana cultivation. The S3 portion of the studied area has severe limitations posed by slope, texture, pH, and depth. The N2 portion of the area is permanently not suitable because of severe limitations posed by slope, depth, texture and soil erosion.

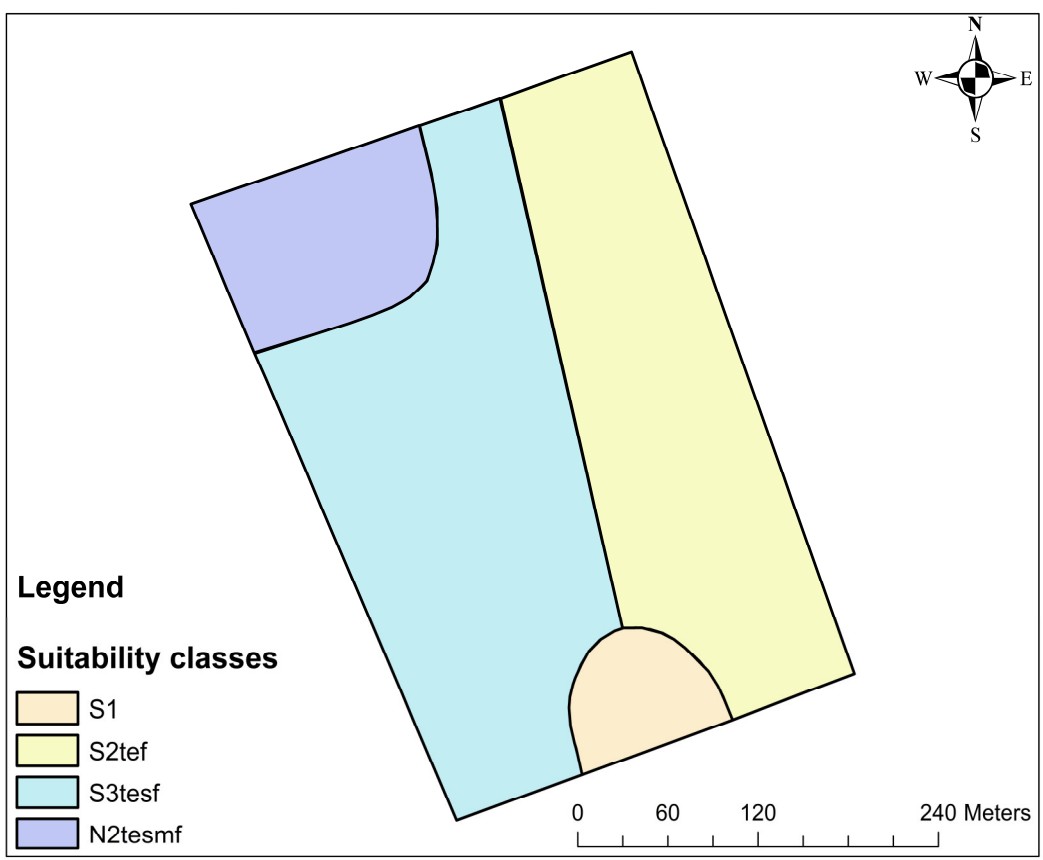

**Figure 4.** Spatial distribution of soil suitability classes across the 12 ha banana plantation. The main liming factors, indicated by the suitability subclasses alongside the suitability classes on the map: t, topography; e, erosion; m, moisture; s, soil physical characteristics (i.e., clay and effective rooting depth); f, soil fertility limitation (i.e., organic carbon and pH).

## 4. Discussion

In this study, we found that a greater proportion of the farm was arable (78%). This arable portion of the farmland was characterised by Westleigh, Hutton and Valsrivier soil forms. These soils were found to be highly (Westleigh), moderately (Hutton) and marginally suitable (Valsrivier-footslope) for banana production. Even though Westleigh soil was arable and highly suitable for banana, the presence of the soft plinthites (an iron rich, humus poor mixture of kaolinitic clay with quartz) below the topsoil horizon may impose limitations [32]. Soils characterised by a clay fraction dominated by kaolinite have a low cation exchange capacity (CEC). The implications of soils with low CEC are that they are likely to develop deficiencies in K, Mg, sulphur (S) and other cations [33]. Another limitation associated with plinthic horizons is that even when soft they do not appear to be well colonised by roots. This is because the plinthites are cemented by iron to the extent that the dry fragments do not soak in water, and cannot be penetrated by roots [34]. The consequence of such cementations is that during a period of heavy rainfall and irrigation, the soil gives rise to waterlogging conditions. Intermittent wetness in the soil directly affects plant growth through oxygen deficiency and indirectly by reducing the availability of nitrogen (N) and sometimes causing manganese (Mn) toxicity [35]. In sandy soils, excessive water can leach nitrate N beyond the rooting zone of the developing plant. In heavier soils (soils with very high clay content), nitrate N can be lost through denitrification (the microbial process of reducing nitrate and nitrite to gaseous forms of N, principally nitrous oxide ($N_2O$) and nitrogen gas ($N_2$) [35].

In as much as the Hutton soil was arable and moderately suitable, it was found to be limited by its location in the field as evident in slope ranging from 4–8%, low levels of OC

(1.1%), and an acidic pH (5.1). Moreover, the Hutton soil had a high clay content (42%) and falls under the Oxisols, which are mostly rich in oxides [33]. In oxides or oxide layer silicate coated systems, phosphorus (P) fixation increases with an increase in clay content [36]. Therefore, the nutrient P in Hutton may be fixed in the soil and not be available for plant uptake.

The land capability assessment further revealed that some portions of the farmland had low arable potential (6%) and 17% of the land was considered non-arable. The low arable potential and non-arable portion of the farmland was characterised by Valsrivier (middleslope) and Glenrosa soil forms respectively. Notably, Valsrivier was found to be marginally suitable while Glenrosa was permanently not suitable for banana production. Limiting factors for the Valsrivier soil included its shallow rooting depth (0.5 m), acidic pH (5.0), low OC (1.54%), and the fact that it was located on steeper slope gradient (4–8%). Similarly, Glenrosa was limited by its location on a steep slope gradient (4–8%), shallow effective rooting depth (0.05 m), low OC (1.1%), lower clay content (21%) compared to the other soil forms and acidic pH (4.6).

In order to grow bananas optimally, the slope percentage must be in the range of 0–3%, clay content ranging from 30% to 50%, pH varying between 5.5 and 7, OC and depth of at least 1.5% and 0.51 m respectively [15,29,37]. At high altitude, banana plants may break since they are prone to wind damage because of their height [37]. Additionally, water tends to travel from less level areas to flat ones during periods of high rainfall or irrigation. This leads to the removal of smaller topsoil particles, which causes soil erosion and subsequent loss of nutrients (they are carried away with finer topsoil). Banana in less flat lands would thus receive less water and nutrients. Moreover, low clay content soils typically have poor capacity to hold water and nutrients. This is explained by the combination of high surface area and density of clay, which causes moisture and nutrients to be retained [35]. The particles that make up clay soil are negatively charged, which means they attract and hold positively charged particles, such Ca, K, and Mg [38]. For these reasons, bananas grown on low clay soils will suffer from water stress and low nutrient availability which would lead to poor crop growth and subsequent yield reduction.

Plant growth and most soil processes, including nutrient availability and microbial activity, are favoured by a soil pH range of 5.5–8 [39]. When soil pH drops, aluminium (Al) becomes soluble. A small drop in pH can result in a large increase in soluble Al [40]. In this form, Al retards root growth, thus restricting access to water and nutrients. Accordingly, poor banana growth and yield reduction would occur as a result of inaccessible water and nutrients [40]. In very acidic soils, all the major plant nutrients (e.g., N, P, K, S, Ca, Mn) and also the trace element molybdenum (Mo) may be unavailable, or only available in insufficient quantities [40]. This is because most microbial processes, including the breakdown of organic matter and cycling of nutrients, are reduced in acidic soil because growth and reproduction of the soil microbes, primarily bacteria and fungi, are reduced [38]. This would explain why there is low OC in such soils even though the farmer practices organic mulching using banana leaf litter. Consequently, this would imply that the soil might not have sufficient nutrients and inadequate water for optimum banana production since OC is responsible for nutrients and water retention.

The defining characteristic of Valsrivier soils is clay enrichment in the subsoil [33] which causes the development of strong structure in the B horizon (Pedocutanic B). The overriding feature of Glenrosa soils is their clear resemblance with the underlying parent rock (Lithocutanic B) [33]. The B horizons (of Valsrivier and Glenrosa) are often sufficiently hard and dense, and as such are an impediment to both root growth and water movement [33]. As a result, the roots of the bananas planted on these soils will remain confined to a small volume of soil that cannot provide adequate anchorage, water and nutrients [41]. Shallow Lithocutanic and Pedocutanic B horizons reduce the usable soil depth and enhance the tendency of soil to waterlogging in heavy rains, and fall below the permanent wilting percentage under drought conditions [42]. Consequently, bananas

grown on these soils will suffer from stunted root growth and waterlogging. Stagnant water in banana farmlands might cause diseases such as the Panama disease (a wilting disease caused by the fungus *Fusarium oxysporuf. sp. Cubense*) [37,43]. This disease can ultimately kill the banana plant in not properly managed. The Panama disease is caused by an upsurge (favoured by reducing soil environment caused by stagnant water) of the solubility and bioavailability of redox-sensitive micronutrients [44]. Increased micronutrient bioavailability from reduced pockets within the crop root zone has been linked to increased *F. oxysporum* pathogenicity [45]. Furthermore, a reducing environment inhibits nitrification, increasing the concentration of soil ammonium, which is favourable to Fusarium wilt development [44].

The excess water in the root zone is accompanied by anaerobic conditions (refer to when the soil has little to no available oxygen) [45]. In the case of plants, oxygen ($O_2$) is a necessary component in many processes including respiration and nutrient movement from the soil into the roots [38]. In the absence of $O_2$, root respiration and nutrient movement are hampered. This is because root respiration in aerobic conditions requires a continuous supply of $O_2$ to the rhizosphere [46]. As a result, the banana plant will show reduced water consumption and stomatal conductance, slow growth, wilting and decreased yield [47,48].

The principal soil forming process of Glenrosa soils is the dissolution and subsequent removal of carbonates [33]. This intensive removal of soil carbonates leads to further eco-logical consequences, mostly related to a decline of soil functions such as decreased net primary production and lower soil organic matter (OM) stability [49]. Soil OM has both a direct (It serves as a source of N, P, S through its mineralization by soil microorganisms) and indirect (is required as an energy source for N-fixing bacteria hence influences the supply of nutrient from other sources) effect on the availability of nutrients for plant growth [50]. Moreover, OM leads to the synthesis of complex organic compounds (e.g., humic and fluvic acids) that bind soil particles into structural units called aggregates [51]. Therefore, the less stable soil OM will contribute to decreased nutrients and a poorly structured soil which would limit water infiltration because of compaction subsequently leading to less water in the root zone [51]. Consequently, bananas grown on these soils will suffer from inadequate water and nutrient supply.

## 5. Conclusions

In conclusion, four soil forms were identified and classified in the study area, namely Hutton, Westleigh, Valsrivier and Glenrosa. The land capability assessment revealed that Makuleke farm is categorised by four land capability classes with class I, III, IV and VI occupying 17%, 61%, 6% and 16% sequentially. In essence, 78% of the farm was arable, 6% had low arable potential while 16% was non arable. Furthermore, soil site suitability assessment revealed that the suitability of the soils at Makuleke farm for banana production is highly variable. The farm was grouped into four suitability classes for banana production; S1 (highly suitable), S2 (moderately suitable), S3 (marginally suitable) and N2 (not suitable), which covered 12%, 34%, 38% and 16% respectively.

Owing to that the farmers at Makuleke were utilising the land and soil resource without prior land evaluation, this contributed to below par banana yield and soil degradation in some portions of the farm. The findings of this study will be useful to decision making and planning at the farm going forward. The land capability and soil suitability assessment of this farm would help to define best agricultural practices to adopt in order to preserve soil functions (soil and water retention). It will help farmers to tailor their soil management practices to specific areas in the farm in order to improve the productivity of the land. By doing so, the farmers will be able to improve banana yield which was affected by a lack of soil information in their plantation.

This study provides baseline for agricultural land assessment. It will help farmers and decision makers in other agroecological zones on how best to conduct land evaluation

in order to improve their agricultural productivity and to avoid inappropriate agricultural practices which might lead to land degradation.

**Author Contributions:** Conceptualization, S.M.S. and P.E.D.; Introduction, S.M.S.; methodology, Results and Discussions, S.M.S. and P.E.D.; S.M.S. and P.E.D.; Reviewing and conclusions, S.M.S. and P.E.D.; writing—review and editing, S.M.S. and P.E.D.; supervision, P.E.D. Funding acquisition, S.M.S. All authors have read and agreed to the published version of the manuscript.

**Funding:** This research was funded by the National Research Foundation (NRF) South Africa, and the grant number is MND210826635262.

**Institutional Review Board Statement:** Not applicable.

**Informed Consent Statement:** Not applicable.

**Data Availability Statement:** Not applicable.

**Acknowledgments:** I would like to thank Mbanjwa V.E. for assisting with soil survey, classification, mapping and with the laboratory analysis of the physical and chemical properties of the collected samples of the study area. I would also like to thank Chauke R. with site selection, soil survey and classification of the soils of the study area. Lastly, I would like to thank Maluleke H (land manager at Makuleke farm) for giving us a permission to conduct soil survey, classification and land evaluation of the farm.

**Conflicts of Interest:** The authors declare no conflict of interest.

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
