# Peer review of "Unlocking the Land Capability and Soil Suitability of Makuleke Farm for Sustainable Banana Production"

_sustainability, doi:10.3390/su15010453_

Round 1
Reviewer 1 Report
Overall the manuscript has presented good work in mapping and describing the findings of the topic discussed. However, the manuscript can be presented, but need to be improved in some issue, see the attached file suggests.

Reviewer 2 Report
There is a huge need for reliable soil information to support agricultural decision-making in smallholder farms to ensure sustainable agricultural production. In this study, the authors survey and quantified the soils properties at Makuleke farms to map land capability and suitability classes. While it is an interesting topic, there are a few issues I’d like the authors to clarify, thus I give major revision. The comments are as follows:
1. In abstract and introduction, the authors haven’t point out the difference of this study with previous studies that also survey and classify the soils. This should be added in both the abstract and introduction to give the readers a better idea of the creativity that this study stresses.
2. The classification standard seems to be pretty subjective in that there is no quantitative standard point out for the “highly suitable”, “moderately suitable”, etc. It would be better if this is point out and so we can have a better idea of the standards, and how this can be applied to actual agricultural production.
3. How much of production yield would be affected by more suitable survey and application of the results? This would be of interest on how impactful this study is to the readers.
4. The study is based on a site specific survey, however, how are the results conclusive to other sites around the globe? It would be good to add one paragraph or two that would demonstrate how this study’s conclusion is impactful over other region to increase its potential impact.
Reviewer 4 Report
The land capability and soil suitability of Makuleke farm were classified and mapped in this study. The findings of this study will be useful to decision making and planning at the farm going forward. It will help farmers to tailor their soil management practices to specific areas in the farm in order to improve the productivity of the land. It's a good paper, while I still have some questions or suggestions about it:
1. In table 1, Soil site criteria for banana fruit includes "Soil fertility characteristics", yet I did not see any test for it in this study.
2. Please plot the sample pits in the map.
3. Please present the Soil organic carbon, pH, EC and other Physicochemical parameters of the soil samples.
4. Please note the "TSD" and ERD for table 2.
5. line 151: What is "I" in the third row and second column of table 1 mean?
6. line 110: by [20].
Round 2
Reviewer 1 Report
Thank you authors, for addressing all my questions and adding more information as requested.
Reviewer 2 Report
The authors have answered my questions, thus I recommend publication.
Reviewer 4 Report
The authors have responded to all the questions I mentioned in the first round. No more comments.